# Mimicking actions is a good strategy for beginners: Fast Reinforcement Learning with few Expert Action Sequences

## Abstract

Imitation Learning is the task of mimicking the behavior of an expert player in a Reinforcement Learning (RL) Environment to enhance the training of a fresh agent (called novice) beginning from scratch. Most of the Reinforcement Learning environments are stochastic in nature, i.e., the state sequences that an agent may encounter usually follow a Markov Decision Process (MDP). This makes the task of mimicking difficult as it is very unlikely that a new agent may encounter same or similar state sequences as an expert. Prior research in Imitation Learning proposes various ways to learn a mapping between the states encountered and the respective actions taken by the expert while mostly being agnostic to the order in which these were performed. Most of these methods need considerable number of states-action pairs to achieve good results. We propose a simple alternative to Imitation Learning by appending the novice's action space with the frequent short action sequences that the expert has taken. This simple modification, surprisingly improves the exploration and significantly outperforms alternative approaches like Dataset Aggregation. We experiment with several popular Atari games and show significant and consistent growth in the score that the new agents achieve using just a few expert action sequences.

## 1 Introduction

Traditional Reinforcement Learning algorithms have a common trait; train from scratch by giving the agent large number of states and responses to actions taken by the agent in a given task/environment. The agent then figures out a way to improve its performance on the task after sufficient time is given. On the contrary, human beings have substantial prior information about the task being performed. For example, in a video game, humans do not take random actions even when we play for the first time because we have some context about the game and goals. In reasonable time, we become adept at playing the particular game. This motivates the idea of Imitation Learning ((Abbeel & Ng, 2004; Argall et al., 2009; Pomerleau, 1989)); the task of augmenting a novice agent (just beginning to train) with the information from an expert. In a typical Imitation Learning set up, the policy of expert is given as an oracle and the novice agent tries to map the states encountered to the actions suggested by the expert.

### 1.1 Challenges of Imitation Learning

Prior work in Imitation Learning demands both state and action information of the expert. Often, this requirement becomes unreasonable for applications like learning to play Atari games (Mnih et al., 2013) because having access to all video frames and the respective actions taken by an expert is infeasible and causes memory overheads. For example, in the case of Atari games, each input frame is 210*160*3 dimensional matrix and the length of an episode of a game could range in few thousands. If we keep accumulating such episodes, we quickly run out of memory as we have several thousands of images and corresponding actions. Also, popular methods need careful and task specific feature engineering (Duan et al., 2017) which is far from desirable. Ideally, we would like to learn from minimal demonstrations of an expert.

Our main contribution is the observation and corroboration of the fact that appending the action space of a novice agent with the most frequent action sequences of an expert helps achieve considerably better exploration, and hence better learning, in less time. Our biggest strength is that, we only need very few reliable trajectories of actions that an expert has taken (about 20 action sequences for ATARI games as we see later in section4; we don't even need the states). This causes negligible memory overhead and we show consistent improvement in scores achieved on popular Atari games. One may draw parallels between our approach and 'Options Framework' for finding extended actions(Sutton et al., 1999; Fox et al., 2017). 'Options' is a hierarchical framework that has multi-level actions like a tree and an agent chooses a path. Our proposal differs from this in the fact that we are not discovering best 'Options', rather we are giving the agent some information about potentially good 'Options' to explore better.

We would like to emphasize that our improvement is time-wise unlike many other approaches that compare episode-wise. Improved performance on episodes with a considerable time overhead for each episode can give a misleading and impractical impression of superiority.

## 2    PREVIOUS WORK

One of the prime works in Imitation Learning is 'Dataset Aggregation (DAgger)'(Ross et al., 2011). It proposes to use a blend of policies of the expert $\pi^*$ and new training agent $\hat{\pi}_i$. At iteration $i$, we act according to the policy

$$\pi_i = \beta_i \pi^* + (1 - \beta_i)\hat{\pi}_i \qquad (1)$$

We collect a sequence of $T$ states and perform the actions given by eqn. 1. At each of these states, the expert policy may give different action to what eqn. 1 suggests. The principle is to minimize this difference between the policies of the oracle and the current agent. DAgger proposes a simple supervised classifier to minimize this 'error' in prediction.

As mentioned earlier, the novice only has access to the policy of the expert as an oracle. This may lead to a wide gap between the policy functions of both the trainee and trainer/expert. For example, if the network architecture of the trainee is too small compared to the expert (whose network is not accessible to the trainee), the intrinsic learning capacity of the trainee is limited. If we force it to learn the map between states and expert suggested actions, the learning may converge poorly.

(He et al., 2012) proposes to alleviate this by demonstrating actions that are not necessarily the best ones but good enough for a given state and are easier for the trainee to achieve. The expert's policy gives a non-zero probability for all possible actions for a given state. The novice's goal would be to predict an action that lowers the classification loss (in principle, it has to be the best action but predicting the second best may not be all that worse). The higher an action is predicted by the novice's policy, the easier it is for it to achieve. The lower the difference between the novice's probabilities and the expert's probabilities, the better it is. Hence, by combining task loss and novice's predictions, (He et al., 2012) comes up with a new policy give by

$$\tilde{\pi}_i(s) = argmax_{a \in A}\lambda_i.score_{\pi_i}(s, a) - L(s, a)$$

where $L(s, a)$ is a loss function for the experts predictions and is chosen to be in $[0, 1]$.

Both the previous approaches need the information about the trajectories traversed (or the actions taken) by the expert to train a classifier in a supervised manner. (Stadie et al., 2017) proposes a method for 'Third-person Imitation Learning' which is to train the current trainee in an unsupervised manner by observing an expert perform a task and inferring that there is a correspondence to itself. In this problem, we are given two MDPs $M_{\pi_E}$ (of an expert) and $M_{\pi_\theta}$ (of novice). Suppose we are given a set of trajectories $\rho = (s_1, ...., s_n)_{i=0}^n$ generated by $\pi_E$ which is optimal under the reward scheme $R_{\pi_\theta}$, the goal is to recover proxy policy $\pi_\theta$ from $\rho$ which is optimal w.r.t the reward scheme $R_{\pi_\theta}$. This work addresses an important concern of learning from people whom you cannot exactly match, i.e., we need not provide with sequences and actions that the novice should have taken. Rather, we can learn from the demonstrations of any person whose reward scheme does not match the novice's but lies in the same environment.

In a recent development, Generative Adversarial Imitation Learning (GAIL (Ho & Ermon, 2016)) proposes a remarkably new approach of model free Imitation Learning. GAIL proposes to reduce the difference between the expert and novice distributions of state-action pairs $D_E(s, a)$ and $D_\theta(s, a)$

respectively. This is different from previous approaches because we don't need to train a supervised classifier. We only need to sample a few state-action pairs generated by novice and expert policies separately. Then, we optimize an information theoretic loss function called Jensen-Shannon divergence that enforces high mutual information between distributions of both novice and expert. It is given by:

$$min_\pi max_D \mathbb{E}_\pi[logD(s,a)] + \mathbb{E}_{\pi_E}[log(1 - D(s,a))] - \lambda H(\pi)$$

Here, $\pi$ and $\pi_E$ are novice and expert policies and novice wants to mimic the behaviour of expert. $H$ is the entropy of novice policy and serves as a regularizer. In a followup to GAIL, Information Maximizing Generative Adversarial Imitation Learning (InfoGAIL (Li et al., 2017)) was proposed to deal with the case of having demonstrations from a mixture of experts as opposed to a single expert. It infers the latent structure of human demonstrations while ignoring noise and variability in various demonstrations by various people.

While the primary advantage of GAIL/InfoGAIL approach is being model-free and unsupervised, it is hard to have a parallel setup where multiple agents play according to a central policy and periodically update the central policy like in the case of Asynchronous Advantage Actor-Critic (A3C,(Mnih et al., 2016)). This causes scalability issues and it is hard to port this algorithm to Atari games. Having parallel threads for GAIL is an orthogonal research work and is beyond the scope of this paper.

Our work differs from the aforementioned approaches in the fact that we do not need any information about the state-action correspondence for the expert. We only have to log actions that an expert takes and identify the frequent sequences as potentially good options for the novice to explore. In our experiments (section 4), we realize that the distribution of action-pairs saturates with access to just as few as 25 episodes.

## 3  OUR PROPOSAL

On a broad note, our proposal is to extend the action set of a Reinforcement Learning (RL) agent by including a small set of $k$-step sequences that an expert in the same environment has frequently taken. For example, let's say an environment has just $N = 2$ actions $\mathcal{A} = \{L, R\}$. All possible $k = 2$-step action sequences are $\mathcal{A}^2 = \{LR, RL, RR, LL\}$.

Suppose we have access to 3 episodes of an expert E's actions: sequence1: $L - R - R - R - L - L - L - L - R - L$, sequence2: $L - L - R - R - R - L - R - R - L - R$ and sequence3: $L - L - L - R - L - R - R - L - R - R$. We accumulate the frequency counts for each action pair and obtain the histogram $H_2 = \{LL : 3 + 1 + 2 = 6, LR : 2 + 3 + 3 = 8, RL : 2 + 2 + 2 = 6, RR : 2 + 3 + 2 = 7\}$. The top $N = 2$ meta actions are $LR$ and $RR$. We now enlarge the action set for a novice agent as $\mathcal{A}^+ = \mathcal{A} \bigcup \{LR, RR\} = \{L, R, LR, RR\}$. We treat all 4 actions as independent. Similarly, if we were to choose top 3-step action sequences, we would obtain the histogram $H_3 = \{LLL : 2+0+1 = 3, LLR : 1+1+1 = 3, LRL : 1+0+1 = 2, RLL : 1+0+0 = 1, LRR : 1 + 2 + 2 = 5, RLR : 0 + 2 + 2 = 4, RRL : 1 + 2 + 1 = 4, RRR : 1 + 1 + 0 = 2\}$. The top-$N$ 3-step actions would then be $\{LRR, RLR/RRL\}$.

We notice that as $k$ increases, picking top-$k$ action sequences becomes harder due to ties (like $RLR$ and $RRL$). As we see in our experiments in section 4, $k = 2$ seems to be an appropriate choice for ATARI games. Please note that we only pick top-$N$ action sequences for any $k$ as we want the growth of effective action space $\mathcal{A}^+$ to be linear in $k, N$. We can then train any popular RL algorithm on $\mathcal{A}^+$ for the novice agent. In our case, we chose to use the state-of-the-art GPU enabled Asynchronous Advantage Actor-Critic (GA3C, (Babaeizadeh et al., 2017)). GA3C essentially has a similar structure to A3C (Mnih et al., 2016). In this set-up, multiple agents play individual games simultaneously with a common central policy and send gradient updates to a central server. The server periodically applies the gradient updates from all the agents and sends out a fresh policy to all the agents. This approach is heavily parallelizable and brings down the training time of RL agents from days to hours.

### 3.1  MOTIVATION:

The primary motivation to pursue this approach comes from the fact that a human baby does not try to infer the utility of the demonstrator while trying to explore the world, but he/she merely tries to replicate frequently performed actions taken by the demonstrator for exploring the world.

Analogously, given a very limited demonstration, trying to learn a mapping between state action pairs is implicitly trying to learn the utility function of the expert. It is too ambitious to learn a reliable mapping by training complex neural network with very few state-action pairs. Instead, during the exploration phase, we trust the observed action sequence rather than our ability to figure out the inner state-action mapping mechanism of the expert.

Consider a novice tennis player who wants to imitate the playing style of an expert player like Roger Federer. He/She could identify Federer's popular mini-moves (action sequences) and try to mimic them as and when possible. A forehand shot whenever possible will involve bending legs, swinging arm, and ending the swing with hands all the way over the other shoulder. By trying to mimic this sequence, humans naturally and reflexively explore faster, understand the importance of each step and later even modify expert behaviour with more experience. In the initial stages, it is pointless to figure out why we need to bend legs or swing your arms all the way long after hitting the ball. Inferring utility comes with enough experience. We precisely experiment with this point and show that merely providing short frequent action sequences (combo-action) taken by expert as an option improves the exploration significantly leading to significantly faster learning.

A preliminary evidence to the feasibility of this approach is given in the report(Anonymous, 2018). The authors append all possible action pairs to the the original action space (causing it to grow exponentially). They devise a method to update both individual actions and their pairs from the same episode thereby extracting more gradients by playing the same number of episodes. In our case, by trimming down the effective action space from exponential to linear using expert action sequences, we increase the scope of this approach.

## 4 EXPERIMENTS

We validate our approach on 8 Atari-2600 games namely *Atlantis*, *SpaceInvaders*, *Qbert*, *DemonAttack*, *BeamRider*, *TimePilot*, *Asteroids* and *FishingDerby*. The number of basic actions in these games ranges from 4 to 18 (see table 4.2). Atari-2600 games are standard benchmarks for Reinforcement Learning(Mnih et al., 2013; 2015; 2016). We program using Tensorflow framework and use the environments provided by OpenAI Gym. We compare our results time-wise against the state-of-the-art algorithm GPU enabled Asynchronous Advantage Actor Critic (GA3C) from NVIDIA whose code is publicly available at `https://github.com/NVlabs/GA3C`. GA3C was carefully developed and curated to make good use of GPU and is the best performing algorithm on Open AI Atari games to the best of our knowledge. We also compare against the popular DAgger algorithm for Imitation Learning.

**Dagger:** Pseudocode for our implementation of DAgger is given in Algorithm 1. It trains a classifier network by obtaining training data from expert's policy and acts according to a mixed policy. As mentioned earlier, DAgger has a major problem of memory explosion in the case of Atari games because the input video frame size is $210 \times 160 \times 3$ and we play thousands of episodes each with number of steps ranging from $790 - 7400$(given in table 4.2); if we keep on appending trajectories indefinitely to the training dataset $D$. For this reason, we limit size of the dataset $D$ to 20, i.e., we only store the last 20 trajectories of states that novice takes as per the blended policy and the respective actions that the expert suggests.

But giving indefinite access to expert's policy makes it an unfair comparison as the whole premise of our approach is the case where we only have very little information about the expert. Hence we chose to limit the number of episodes for which expert policy is available. We show results for different episode limits- 100, 500 and 1000 to identify how many episodes are needed to give a good start that can match our proposal in the longer run. After the limit is reached, access to expert policy is taken away and the residual network of the novice continues to train using the usual GA3C algorithm until we complete a total 15 hrs training time (choice of 15 hrs is explained in section 4.1).

**InfoGAIL:** We also tried to compare our approach to InfoGAIL. We used the publicly available code for InfoGAIL and made only a few essential changes to use it for Atari-2600 games. InfoGAIL uses auxiliary information like velocity and acceleration of vehicle along with the current video frame. Since we do not have such information in Atari games, we stack up the latest 4 video frames. To be more precise, at each state, we have a $210 \times 160$ image. We first take a grey scale version of the image using the popular $np.dot(rgb[..., :3], [0.299, 0.587, 0.114])$ step. We then resize the

image to $84 * 84$. We then concatenate the previous 3 $84 * 84$ frames with the current one to obtain a $84 * 84 * 4$ input to the neural network. Please note that we used this pre-processing for our method, GA3C, Dagger and InfoGAIL for a fair comparison. InfoGAIL also expects features extracted from a pre-trained convolutional neural network trained on ImageNet dataset. We use VGG16 network to extract 512 dimensional features (we do not use ResNet50 because 2048 dimensional features cause memory overheads with larger batch sizes). As for the expert demonstrations, we provided state-action pairs for 100 episodes each with a trajectory length of 100. This amounts to 10000 state action-pairs for each game.

**Challenges with InfoGAIL:** Unfortunately, despite trying with several hyper-parameter settings, we could not get InfoGAIL to converge to the expected scores. The overall scores for an episode seem to either stagnate or even worse diverge in one game (Qbert). As mentioned before, this behaviour is plausible because InfoGAIL does not use multiple parallel threads like A3C. In our experience, we see that getting convergence on Atari games without any parallelization takes a few days to train. Since we limit ourselves to 15 hrs of training, it's hard for a Generative Adversarial setup to converge. Nevertheless, we show the pseudocode in algorithm 2 and time-wise plots in figure 4 for InfoGAIL in the supplementary material.

For a game with $N$ basic actions, we have a total of $N^k$ $k$-step meta-actions. Using all possible $k$-tuples of actions leads to exponentially exploding action space and is infeasible for games with large basic action space. Hence, we choose the top $N$ meta actions taken by the expert among the $N^k$ possibilities. This keeps the effective action space to $kN$ which grows linearly in $k$ and $N$. We limit the size of meta-actions $k$ to 2 because large action spaces may lead to poor convergence.

---

**Algorithm 1** Algorithm for our variant of Dagger

Initialize $D \leftarrow \phi$
Let $\hat{\pi}_i$ denote the novice's policy at iteration $i$
Let $\pi^*$ denote the expert's policy
Let $M$ be the maximum length of $D$      ▷ At most $M$ trajectories are stored in the data buffer $D$
Let $K$ be the maximum episode for which expert policy is available $D$    ▷ After $K$ steps, residual network trains in usual GA3C style without $\pi^*$
Obtain $\hat{\pi}_1$ by randomly initializing novice's network weights
**for** each parallel thread (as in A3C) **do**
    **for** $i = 1 : K$ **do**
        Let $\pi_i = \beta_i \pi^* + (1 - \beta_i)\hat{\pi}_i$
        Sample $T$-step trajectory by playing using $\pi_i$
        Get dataset $D_i = (s, \pi^*(s))$ where $s$ is a state visited in the $T$-step trajectory
        Append $D_i$ to $D : D \leftarrow D \cup D_i$

        **if** $len(D) > M$ **then** $D \leftarrow D[-M :]$
        **end if**
        Train novice's classifier network on supervised data $D$
    **end for**
    **for** $i = K : N$ **do**
        $t \leftarrow 0$
        **repeat**
            Perform $a_t$ according to policy $\hat{\pi}_i$
            Receive reward $r_t$ and new state $s_{t+1}$
            $t \leftarrow t + 1$
        **until** terminal $s_t$
        $R \leftarrow 0$
        **for** $i \in t - 1, ...., t_{start}$ **do**
            $R \leftarrow r_i + \gamma R$
            Optimize novice's network using $(s_t, a_t, R)$ as in A3C
        **end for**
    **end for**
**end for**
**return** $\hat{\pi}_N$

---

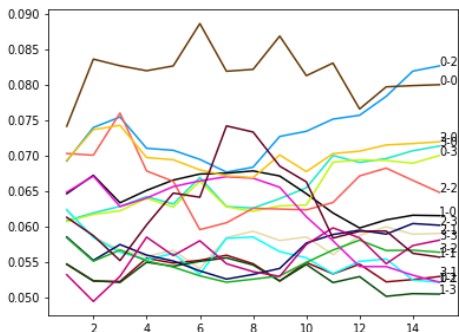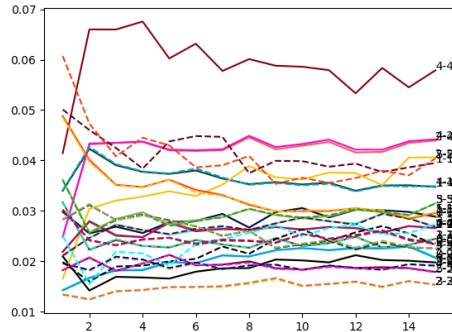

Figure 1: Action Pair distribution for $Atlantis$ (on the left, $4^2 = 16$ pairs) and $SpaceInvaders$(on the right, $6^2 = 36$ pairs)

## 4.1 HOW DO WE GET EXPERT META-ACTIONS?

Our focus is to train a novice agent under the assumption that we have human-expert action sequences. Obtaining human actions is hard because we need to find people that play a variety of Atari games. Hence we chose to treat a neural network that is trained using the state-of-the-art GA3C algorithm as an expert. In the process of training this network, the distribution of action pairs taken by the agent saturates as the average scores per episode converge. We can then identify the top action pairs. The onset of such a saturation is hard to quantify and detect automatically. Hence, we standardize the choice of an 'expert phase' while training GA3C ,i.e., we accumulate the counts for each action pair ($N^2$ such pairs if a game has $N$ basic actions) taken after 15 hrs of training GA3C. To further clarify, we train GA3C for 15 hrs in total. We then freeze this network and call it the 'expert network'. We then play a 'few' ($\sim 25$) episodes with each episode played until termination. We then obtain the histogram of all action pairs and pick the top-N action pairs. The justification for the 15 hr threshold is shown in figure 1. We notice that the distribution of the action pairs does not change significantly after training for about 14 hrs. This suggests the onset of 'expert phase'. Figure 1 only shows the time distribution of meta-actions for 2 games ($Atlantis$ and $SpaceInvaders$ with 4 and 6 actions respectively) to reduce clutter in plots.

Please note that the time for such saturation is dependent on the computing infrastructure. In our case, we use a 16 core CPU with 122 GB RAM and a single Tesla M60 GPU with 8GB memory. Each game plays different number of episodes in the stipulated time. Hence, we provide the approximate number of episodes after which saturation happens for each of the 8 games in table 4.2 to help reproduce results on other machines.

Also, the 15hrs time to train the 'expert network' should not be included in the training time of novice as we are only substituting humans with a trained network. Once we obtain the frequent meta-actions, we begin to train the novice from from scratch with enlarged action space.

## 4.2 NETWORK ARCHITECTURE

Our network architecture is similar to the one used by A3C and GA3C algorithms except for the output layer that has twice the number of actions as opposed to the former ones. We use 2 convolutional layers; first with $8 \times 8$ filters (32 of them) and the second with $4 \times 4$ filters (64 of them). They are followed by a dense layer with 256 nodes. The last layer is the typical softmax layer with as many nodes as the effective number of actions (basic+meta). There is a parallel last layer for Value function similar to A3C. The input video frames are all reshaped to $84 \times 84$.

Table 1: Game Information. Saturation episode refers to the approximate episode after which action pair distribution remains stable.

| Game | Basic Actions | Saturation episode | Avg steps per episode | Top action pairs |
|---|---|---|---|---|
| Atlantis | 4 | 17000 | 7438.38 | 0-2, 0-0, 2-0, 3-0 |
| SpaceInvaders | 6 | 39000 | 1377.24 | 4-4, 4-2, 2-4, 2-2, 1-1, 5-5 |
| Qbert | 6 | 60000 | 1794.18 | 3-3, 2-2, 5-5, 0-0, 1-1, 4-4 |
| DemonAttack | 6 | 32000 | 2645.96 | 4-4, 5-5, 3-3, 1-1, 2-2, 4-2 |
| BeamRider | 9 | 21000 | 3424.68 | 1-1, 8-8, 8-1, 7-7, 7-1,1-8, 4-8, 8-4, 7-8 |
| TimePilot | 10 | 32000 | 1628.03 | 8-8, 0-0, 0-8, 4-8, 8-4,8-0, 4-4, 0-4, 4-0, 1-8 |
| Asteroids | 14 | 63000 | 791.1 | 0-1, 1-0, 1-1, 0-0, 13-0,13-13, 0-13, 1-8, 7-13, 8-1, 8-8, 3-1, 10-10, 1-10 |
| FishingDerby | 18 | 22000 | 1441.09 | 17-17, 13-13, 13-17, 17-13, 9-9, 17-9, 9-17, 5-5, 17-5, 13-9, 9-13, 13-5, 5-17, 12-12, 5-13, 5-9, 12-17, 12-13 |

### 4.3 RESULTS

Figure 2 compares our proposal to GA3C and DAgger (with varying number of expert episodes) for each of the 8 games mentioned before. Each dark line in the plot is the mean of 5 rounds of training. The standard deviation of the curves is plotted in mild color to show how variant each algorithm is for every game. The green curves correspond to our proposed method, red corresponds to GA3C baseline and the rest correspond to Dagger with different limit on access to expert policy. We notice that our method consistently outperforms other baselines by a huge margin in all games. The closest that GA3C or Dagger could get to our scores are on the game FishingDerby. We also observe that giving access to more expert episodes translates to better overall performance of Dagger on games like FishingDerby and Asteroids. In all other games, it's interesting to note that the Dagger curves seem to grow better than GA3C in the beginning of the training but do not sustain the same growth after the objective function is changed. Please note that our method has multi-step actions which could potentially mean that the agent takes more actions per one query to the policy. Hence, in the given time, we play more episodes than other baselines. While we care for time-wise comparison, we have added the episode-wise comparison in the supplementary material 3.

### 5 CONCLUSIONS

We propose a simple alternative to Imitation Learning by just giving a novice trainee the information about popular action-sequences that an expert player takes in an RL environment. We show that our approach outperforms the state-of-the-art GA3C and popular DAgger algorithm for Imitation Learning consistently by significant margin time-wise. We validate our approach on 8 Atari-2600 games. Our approach has no memory and time overheads unlike other methods for Imitation Learning and hence scalable to games with large video frame inputs. There is a huge scope for researching how to learn the best meta-actions without an expert by analyzing the action sequences within a novice's training time. Such a direction would intrinsically align with discovering best 'Options' in 'Options Framework'. We will examine that in our future work.

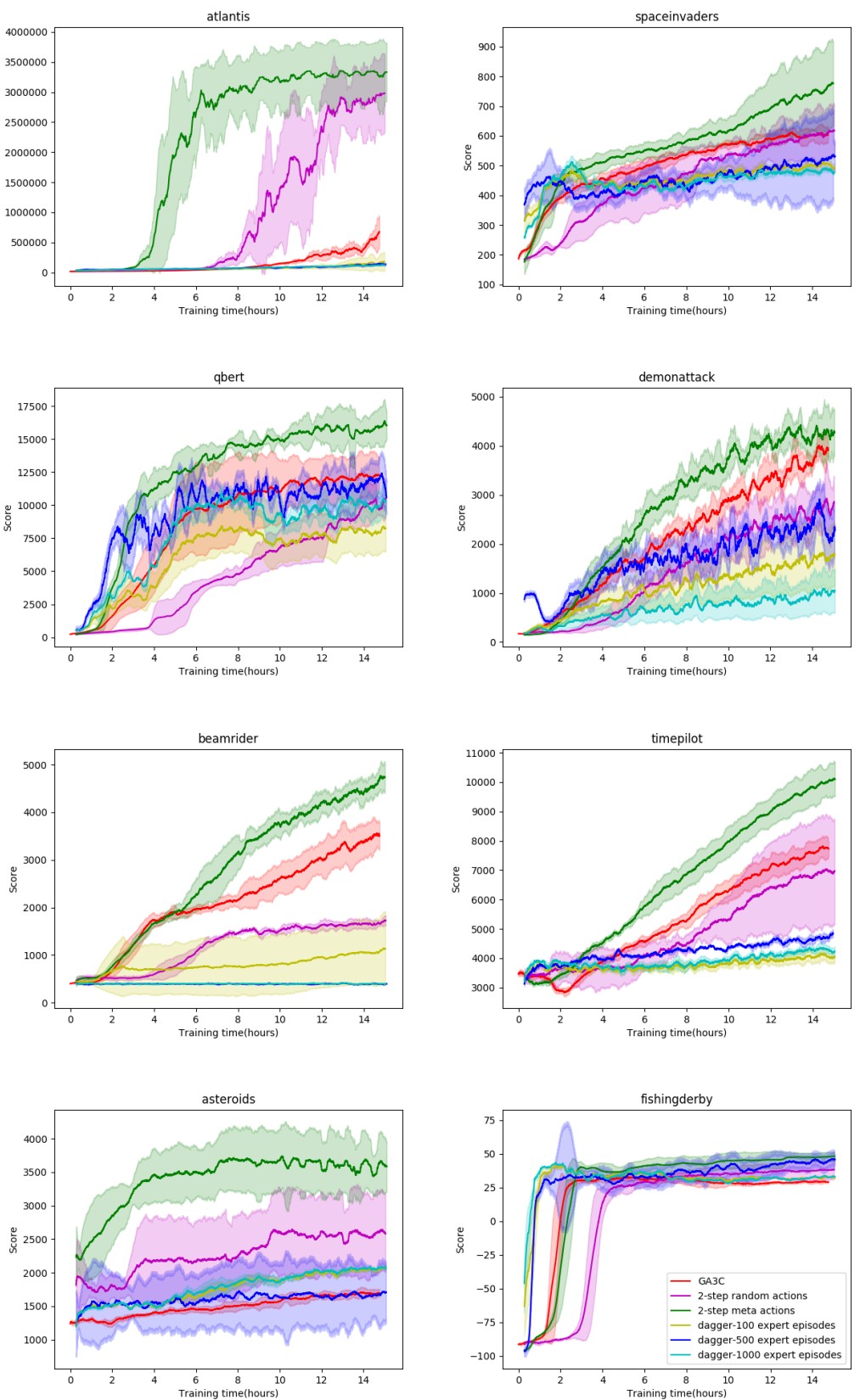

Figure 2: Time-wise comparison of our proposal against GA3C and DAgger. The common legend for all plots is shown in the last game $FishingDerby$.

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

## SUPPLEMENTARY MATERIAL

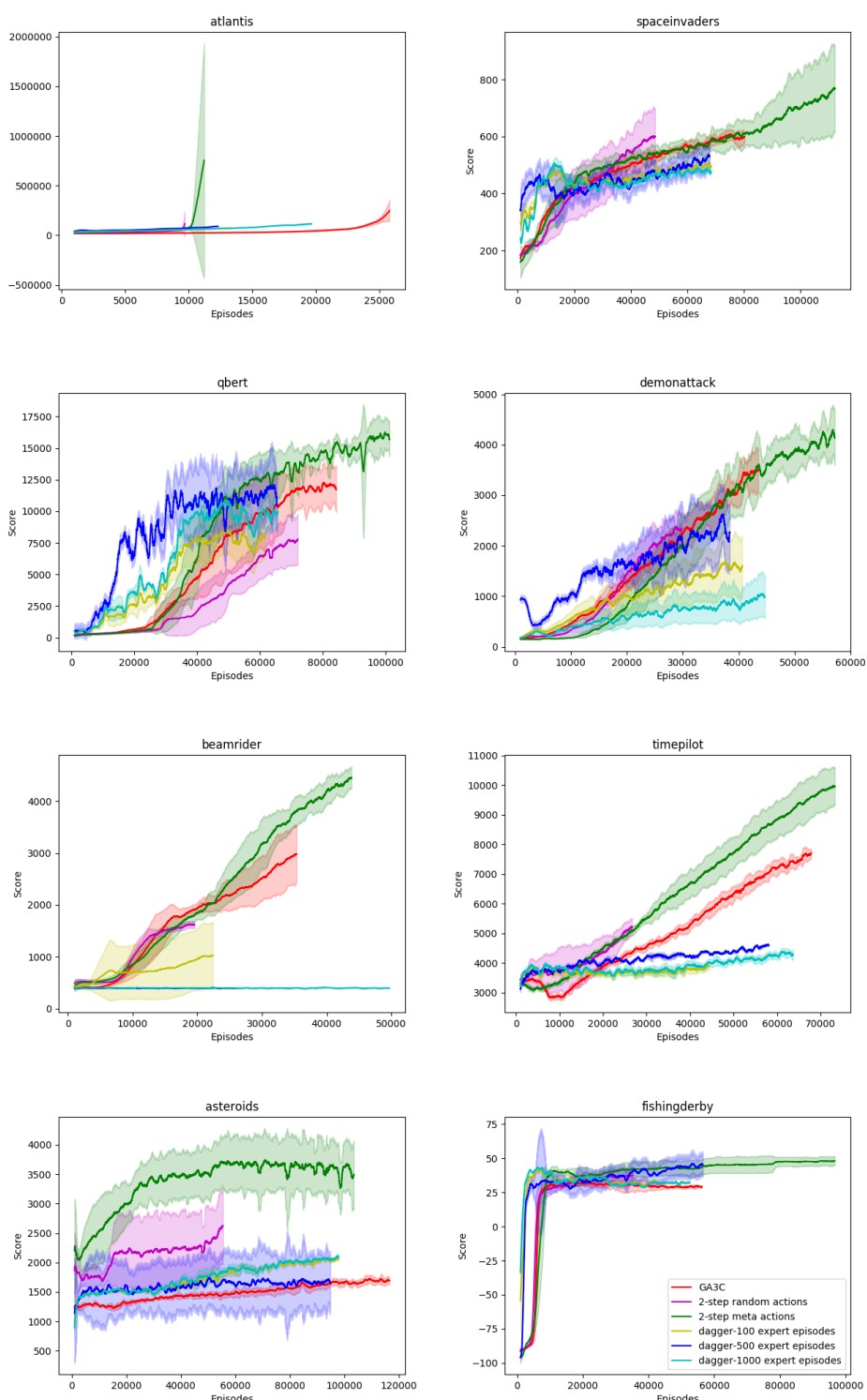

Figure 3: Episode-wise comparison of our proposal against GA3C and DAgger. The common legend for all plots is shown in the last game $FishingDerby$.

**Algorithm 2** Algorithm for InfoGAIL

**Input:** Initial parameters of policy, discriminator and posterior approximation $\theta_0, \omega_0, \psi_0$; expert trajectories $\tau_E \sim \pi_E$
**Output:** Learned policy $\pi_\theta$
**for** $i = 0, 1, 2, ...$ **do**
    Sample batch of latent codes: $c_i \sim p(c)$
    Sample trajectories: $\tau_i \sim \pi_{\theta_i}(c_i)$, with the latent code fixed during each rollout.
    Sample state-action pairs $\chi_i \sim \tau_i$ and $\chi_E \sim \tau_E$ with same batch size.
    Update $\omega_i$ to $\omega_{i+1}$ by ascending with gradients

$$\delta_{\omega_i} = \hat{\mathbb{E}}_{\chi_i}[\nabla_{\omega_i} log(D_{\omega_i}(s,a))] + \hat{\mathbb{E}}_{\chi_E}[\nabla_{\omega_i} log(1 - D_{\omega_i}(s,a))]$$

    Update $\psi_i$ to $\psi_{i+1}$ by descending with gradients

$$\delta_{\psi_i} = -\lambda_1 \hat{\mathbb{E}}_{\chi_i}[\nabla_{\psi_i} log(Q_{\psi_i}(c/s,a))]$$

    Take a policy step from $\theta_i$ to $\theta_{i+1}$, using the objective:

$$\hat{\mathbb{E}}_{\chi_i}[log(D_{\omega_{i+1}}(s,a))] - \lambda_1 L_1(\pi_{\theta_i}, Q_{\psi_{i+1}}) - \lambda_2 H(\pi_{\theta_i})$$

**end for**

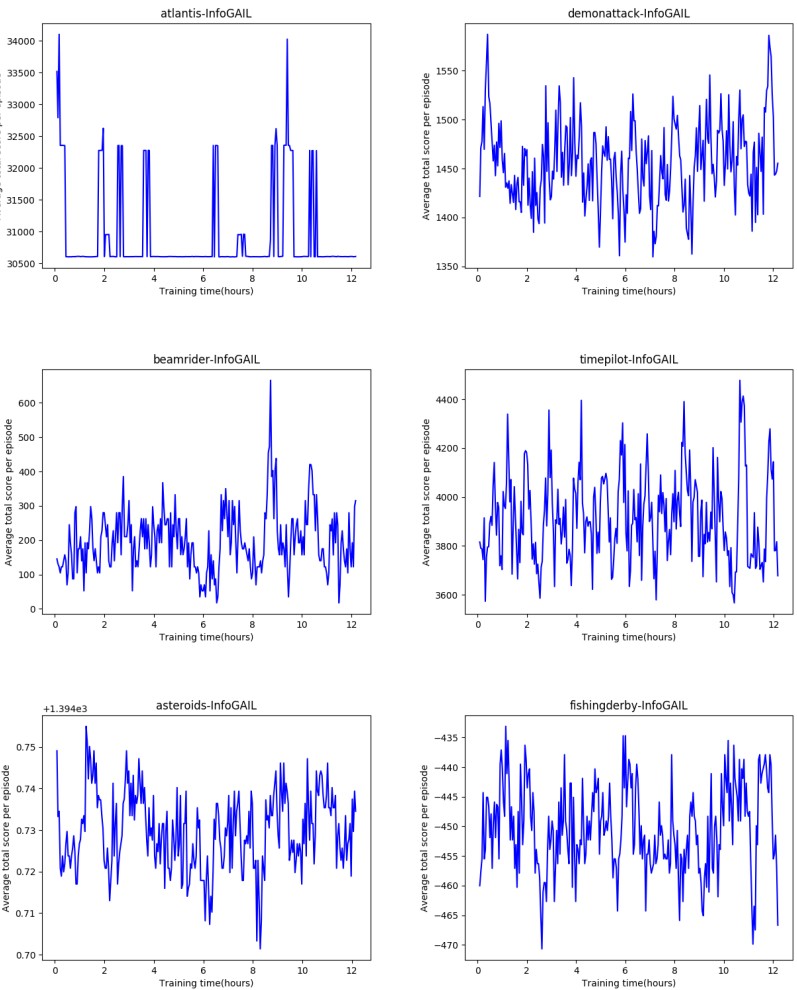

Figure 4: Time-wise comparison InfoGAIL

