# OpenReview forum: "Mimicking actions is a good strategy for beginners: Fast Reinforcement Learning with Expert Action Sequences"
_ICLR.cc/2019/Conference_

### Official Review · AnonReviewer2 · 2018-10-19
**A well written paper with a simple, yet powerfull, idea that needs further analysis**

**Rating:** 6
**Confidence:** 4

**Review:**

The paper describes an imitation reinforcement learning approach where
the primitive actions of the agent are augmented with the most common
sequences of actions perform by experts. It is experimentally shown
how this simple change has clear improvements in the performance of
the system in Atari games. In practice, the authors double the number
of primitive actions with the most frequent double actions perform by
experts.

A positive aspect of this paper comes from the simplicity of the
idea. There are however several issues that should be taken into
account:
- It is not clear how to determine when the distribution of action
  pair saturates. This is relevant for the use of the proposed approach.
- The total training time should consider both the initial time to
  obtain the extra pairs of frequent actions plus the subsequent
  training time used by the system. Either obtained from a learning
  system (15 hours) or by collecting traces of human experts (< 1
  hour?).
- It would be interesting to see the performance of the system with
  all the possible pairs of primitive actions and with a random subset
  of these pairs, to show the benefits of choosing the most frequent
  pairs used by the expert.
- This analysis could be easily extended to triplets and so on, as
  long as they are the most frequently used by experts.
- The inclusion of macro-actions has been extensively studied in
  search algorithms. In general, the utility of those macros depends on
  the effectiveness of the heuristic function. Perhaps the authors
  could revise some of the literature.
- Choosing the most frequent pairs in all the game may not be a
  suitable strategy. Some sequences of actions may be more frequent
  (important) at certain stage of the game (e.g., at the beginning/end
  of the game) and the most frequent sequences over all the game may
  introduce additional noise in those cases.

The paper is well written and easy to follow, there are however, some
small typos:
- expert(whose => expert (whose
% there are several places where there is no space between a word and
% its following right parenthesis
- don't need train => don't need to train
- experiments4. => experiments.
- Atmost => At most

---

> ### Author Response · Authors · 2018-11-27
> **Thank you very much for the positive comments and suggestion about another baseline. Clarifications are given below.**
>
> Please check the updated figures in our paper that include the comparison of a random subset of action pairs vs the most frequent action pairs (the line in magenta). The new plots strengthen our proposal that the most-frequent action pairs have useful information.
>
> Q1. Ideally, an expert should be consistent with the action pair distribution over a set of few episodes. In our analysis, we found that the frequent action pairs after 12 hrs, 13 hrs, 14 hrs and 15 hrs of training the expert network are consistent. Hence, it is evident that after training the expert network for reasonable time, the top action pairs saturate. We have made our choice more concrete by training all expert networks for 15 hrs.
>
> Q2. As mentioned in the paper, imitation learning algorithms presume that the expert information is available beforehand. We just substitute human data with a pre-trained network. Collecting traces of human data is a fast and viable but it is highly dependent on the task/game. In our case, assuming access to expert action sequences, calculating the frequency distribution and obtaining top action sequences is a trivial task with few seconds of time.
>
> Q3. Please check the new plots for random subset of action-pairs. As for adding all possible action-pairs, the action space grows exponentially, and the network must classify lot more classes with the same information. With games like FishingDerby and Asteroids (18 and 14 actions), it become too hard for network to classify hundreds of classes with same information.
>
> Q4. Action triplets are inconsistent and statistically insignificant with limited demonstration: Our focus was on using very limited (small) demonstration. The number of episodes that we use is quite small (25 episodes each with actions ranging from 700 to 7000) as we wanted very limited demonstration. We observe that with such limited demonstration, only action-pairs are reliable. The frequent action triplets after 12 hrs, 13 hrs, 14 hrs and 15 hrs of training expert network are different each time. Furthermore, for the game FishingDerby with 18 basic actions, the top 18 action pairs account for 33.85% of all the action-pairs in the 25 expert episodes. The top 18 action triplets account to just 7.36% of the all triplets in the same 25 episodes. Even for other games, we have similar discrepancy for action pairs vs triplets (DemonAttack-27.21% vs 8.87%, Asteroids-21.67% vs 6.37%, Atlantis 31.32% vs 9.61%, SpaceInvaders-28.82% vs 10.24%, BeamRider-14.78% vs 2.81%, TimePilot-15.41% vs 2.05%, Qbert-67% vs 51%). We still experimented with 3-step actions and noticed that for Atlantis, action triplets outperform action-pairs which Is great. But for other games, action-triplets perform worse than action-pairs.
>
> Q5. Thank you for the suggestion. We’ll investigate search algorithms in the future to identify informative action-sequences. One class of models that we mentioned in the paper is ‘Options Framework’. The main drawback of Options Framework is that we need human designed options. Our work is a generic way of identifying options.
>
> Thank you for spotting typos. We have fixed them in the latest revision.

---

### Official Review · AnonReviewer1 · 2018-10-28
**Border line paper**

**Rating:** 5
**Confidence:** 2

**Review:**

The paper proposes an idea of using the most frequent expert action sequence to assist the novice, which, as claimed, has lower memory overhead than other imitation learning methodologies. The authors present comparison of their proposed method with state-of-the-art and show its superior performance. However I do have the following few questions.

1. The proposed method requires a long time of GA3C training. How is that a fair comparison in Figure 2, where proposed method already has a lead over GA3C? It could be argued that it's not using all of the training outcome, but have the authors considered other form of experts and see how that works?

2. The authors claimed one of the advantages of their method is reducing the memory overhead. Some supporting experiments will be more convincing.

3. In Figure 3, atlantis panel, the score shows huge variance, which is not seen in Figure 2. Are they generated from the same runs? Could the authors give some explanation on the phenomenon in Figure 3?

Overall, I think the paper has an interesting idea. But the above unresolved questions raises some challenge on its credibility and reproducibility.

---

> ### Author Response · Authors · 2018-11-27
> **Thank you for the review. Clarifications are given below.**
>
> Q1. We would like to stress that our setting, also clearly mentioned in the paper at several places, is standard imitation learning setting, where access to expert information is given input to the algorithm. We do not need any GA3C training. It is a proxy to generate very few expert action sequences.  For the other imitation learning baselines, the same pretrained GA3C training is used as a proxy for expert. Hence, it is a fair comparison.
>
> Q2.  The memory advantage of our approach is quite straight forward. Out of all imitation baseline, only our method does not need to store state information at all. We only need few action sequences for ~25 episodes (each with a few 1000 integers) which takes trivially low memory. On the other hand, to store any reasonable (say 10000) state-action pairs of an expert in an environment, we will need at least 4032MB memory. Please note that each state is an image is originally 210*160*3 dimensional.
>
> Q3. As we understand, you’re concerned about difference in variance when we plot episode-wise and time-wise.  Please note that we ran all the 5 runs of each game for 15 hrs. But the number of episodes in each run is different. For Atlantis game, the number of episodes range between 9114 to 10366. In our episode-wise plots, we only show the mean and variance of first 9114 episodes for each run. Hence, even though the time-wise and episode-wise plots are generated from the same output, the variance is higher for episode-wise plot. This is more glaring on Atlantis game as our idea gets much higher score than the baselines.
>
> We believe we have answered all your questions. If you have any questions on reproducibility, we’ve our code ready for release once the review period is over. Since our idea is simple and very effective, it needs more visibility so that more investigation can be made on this idea. Simplicity is the very reason why we can beat GA3C by significant margin. If the idea is not computationally simple, most likely it won’t beat GA3C (a highly optimized implementation on GPUs) on running time.  We hope you will change your opinion about the overall score.

---

### Official Review · AnonReviewer4 · 2018-11-10
**need more in-depth analysis**

**Rating:** 5
**Confidence:** 3

**Review:**

[Summary]

This paper presents an interesting idea that to append the agent's action space with the expert's most frequent action pairs, by which the agent can perform better exploration as to achieve the same performance in a shorter time. The authors show performance gain by comparing their method with two baselines - Dagger and InfoGAIL.


[Stengths]

The proposed method is simple yet effective, and I really like the analogy to mini-moves in sports as per the motivation section.


[Concerns]

- How to choose the number and length of the action sequences?
The authors empirically add the same number of expert's action sequences as the basic ones and select the length k as 2. However, no ablation studies are performed to demonstrate the sensitivity of the selected hyperparameters. Although the authors claim that "we limit the size of meta-actions k to 2 because large action spaces may lead to poor convergence", a more systematic evaluation is needed. How will the performance change if we add more and longer action sequences? When will the performance reach a plateau? How does it vary between different environments?

- Analysis of the selected action sequences.
It might be better to add more analysis of the selected action sequences. What are the most frequent action pairs? How does it differ from game to game? What if the action pairs are selected in a random fashion?

- Justification of the motivation
The major motivation of the method is to release the burden of memory overheads. However, no quantitative evaluations are provided as to justify the claim. Considering that the input images are resized to 84x84, storing them should not be particularly expensive.

- The choice of baseline.
InfoGAIL (Li et al., 2017) is proposed to identify the latent structures in the expert's demonstration, hence it is not clear to me how it suits the tasks in the paper. The paper also lacks details describing how they implemented the baselines, e.g. beta in Dagger and the length of the latent vector in InfoGAIL.

- The authors only show experiments in Atari games, where the action space is discrete. It would be interesting to see if the idea can generalize to continuous action space. Is it possible to cluster the expert action sequences and form some basis for the agent to select?

- Typos
{LRR, RLR/RRL} --> {LRR, RLR, RRL}
sclability --> scalability
we don't need train a ... --> we don't need to train a ...
Atmost --> At most


[Recommendation]

The idea presents in the paper is simple yet seemingly effective. However, the paper lacks a proper evaluation of the proposed method, and I don't think this paper is ready with the current set of experiments. I will decide my final rating based on the authors' response to the above concerns.

---

> ### Author Response · Authors · 2018-11-27
> **Thank you for the review. Clarifications are provided below.**
>
> Q1. Action triplets are inconsistent and statistically insignificant with limited demonstration: Our focus was on using very limited (small) demonstration. The number of episodes that we use is quite small (25 episodes each with actions ranging from 700 to 7000) as we wanted very limited demonstration. We observe that with such limited demonstration, only action-pairs are reliable. An expert should be consistent with the frequent action pairs/triplets over a set of episodes. In our analysis, we found that the frequent action pairs are consistent after 12 hrs, 13 hrs, 14 hrs and 15 hrs of training the expert network. The action pairs at different time instants in training were just permutations of each other. The same was not true for action triplets. Furthermore, for the game FishingDerby with 18 basic actions, the top 18 action pairs account for 33.85% of all the action-pairs in the 25 expert episodes. The top 18 action triplets account to just 7.36% of the all triplets in the same 25 episodes. Even for other games, we have similar discrepancy for action pairs vs triplets (DemonAttack-27.21% vs 8.87%, Asteroids-21.67% vs 6.37%, Atlantis 31.32% vs 9.61%, SpaceInvaders-28.82% vs 10.24%, BeamRider-14.78% vs 2.81%, TimePilot-15.41% vs 2.05%, Qbert-67% vs 51%). We still experimented with 3-step actions and noticed that for Atlantis, action triplets outperform action-pairs which Is great. But for other games, action-triplets perform worse than action-pairs.
>
> Q2. Thank you for the suggestion. It is an interesting exercise to interpret frequent action pairs.
>
> Q3. The memory advantage of our approach is quite straight forward. Out of all imitation baseline, only our method does not need to store state information at all. Even when we are resizing images to 84*84*4, we need several thousands of those images to get noticeable advantage when compared to having no information at all.
>
> Q4. InfoGAIL in one of the most recent techniques in Imitation Learning. Hence, we wanted to compare against InfoGAIL and ensure that we are not missing any subtleties. For our Dagger implementation, we used the simple parameter free version of beta=Indicator(i=1), i.e., 1 for the first episode and then 0 from the second episode.
>
> Q5. Thanks for the great suggestion! We were intending to explore this direction in future on continuous action spaces by binning continuous values to discrete.
>
> Thank you for spotting typos, we have corrected them in the current version. Since our idea is simple and very effective, it needs more visibility so that more investigation can be made on this idea. Simplicity is the very reason why we can beat GA3C by significant margin. If the idea is not computationally simple, most likely it won’t beat GA3C (a highly optimized implementation on GPUs) on running time.  We hope you will change your opinion about the overall score.

---

> > ### Comment · AnonReviewer4 · 2018-12-09
> > **Rating remains the same**
> >
> > Dear authors,
> >
> > Thank you for your clarifications and an additional comparison with a baseline using a random subset of action pairs.
> >
> > The main idea of this paper is interesting. However, my major concern is still in the experimental part, as it remains unclear to me how we should use the method. Many of the hyperparameters are empirically selected and lack a systematic evaluation, e.g. the number and the length of the "meta-actions".
> >
> > Your previous response has shown the performance on action-triplets. It is surprising to me that longer "meta-actions" leads to worse performance, which is somewhat against the main vein of the paper: the training difficulty is not clearly described. It would be better to show the correlation between the number of available demonstrations and the length/number of "meta-actions" we should adopt.
> >
> > I'm still confused about the selection of the baseline. Again, InfoGAIL is proposed to imitate multi-modal expert demonstrations. The tasks used in the paper do not seem to be in this particular setting. GAIL [1] might be a more suitable baseline.
> >
> > As a result, my rating will stay the same for now, but I encourage the authors to keep on improving the paper.
> >
> > [1] Jonathan Ho, Stefano Ermon. "Generative Adversarial Imitation Learning". In NIPS 2016.

---

### Author Response · Authors · 2018-11-19
**Additional experiments and revision**

We have added the plots for another baseline which is to choose a random subset of action pairs and append to the original action space. Please check the new plots in magenta in Figure 2. These plots strengthen our hypothesis that the frequent action pairs have useful information that random action pairs do not.

---

### Meta-Review · Area_Chair1 · 2018-12-14
**Nice work with potential, but contributions need to be strengthened**

**Confidence:** 4
**Recommendation:** Reject

**Metareview:**

The paper proposes an interesting idea for more effective imitation learning.  The idea is to include short actions sequences as labels (in addition to the basic actions) in imitation learning.  Results on a few Atari games demonstrate the potential of this approach.

Reviewers generally like the idea, think it is simple, and are encouraged by its empirical support.  That said, the work still appears somewhat preliminary in the current stage: (1) some reviewer is still in doubt about the chosen baseline; (2) empirical evidence is all in the similar set of Atari games --- how broadly is this approach applicable?